# One Vertex Attack on Graph Neural Networks-based Spatiotemporal Forecasting

## Abstract

Spatiotemporal forecasting plays an essential role in intelligent transportation systems (ITS) and numerous applications, such as route planning, navigation, and automatic driving. Deep Spatiotemporal Graph Neural Networks, which capture both spatial and temporal patterns, have achieved great success in traffic forecasting applications. Though Deep Neural Networks (DNNs) have been proven to be vulnerable to carefully designed perturbations in multiple domains like objection classification and graph classification, these adversarial works cannot be directly applied to spatiotemporal GNNs because of their causality and spatiotemporal mechanism. There is still a lack of studies on the vulnerability and robustness of spatiotemporal GNNs. Particularly, if spatiotemporal GNNs are vulnerable in real-world traffic applications, a hacker can easily cause serious traffic congestion and even a city-scale breakdown. To fill this gap, we design One Vertex Attack to break deep spatiotemporal GNNs by attacking a single one vertex. To achieve this, we apply the genetic algorithm with a universal attack method as the evaluation function to locate the weakest vertex; then perturbations are generated by solving an optimization problem with the inverse estimation. Empirical studies prove that perturbations in one vertex can be diffused into most of the graph when spatiotemporal GNNs are under One Vertex Attack.

## 1 Introduction

Spatiotemporal traffic forecasting has been a long-standing research topic and a fundamental application in intelligent transportation systems (ITS). For instance, with better prediction of future traffic states, navigation apps can help drivers avoid traffic congestion, and traffic signals can manage traffic flows to increase network capacity. Essentially, traffic forecasting can be modeled as a multivariate time series prediction problem for a network of connected sensors based on the topology of road networks. Given the complex spatial and temporal patterns governed by traffic dynamics and road network structure (Roddick & Spiliopoulou, 1999), recent studies have developed various Graph Neural Networks-based traffic forecasting models (Yu et al., 2018; Wu et al., 2019; Li et al., 2017; Guo et al., 2019).

These deep learning models have achieved superior performance compared with traditional multivariate time series forecasting models such as vector autoregression (VAR). However, recent research has shown that deep learning frameworks are very vulnerable to carefully designed attacks (Kurakin et al., 2016b; Goodfellow et al., 2014; Papernot et al., 2016a; Tramèr et al., 2017; Kurakin et al., 2016a). This raises a critical concern about the application of spatiotemporal GNN-based models for real-world traffic forecasting, in which robustness and reliability are of ultimate importance.

For example, with a vulnerable forecasting model, a hacker can manipulate the predicted traffic states. Feeding these manipulated values into the downstream application can cause severe problems such as traffic congestion and even city-scale breakdown. However, it remains unclear how vulnerable these GNN-based spatiotemporal forecasting models are. Particularly, previous adversarial works cannot be directly applied to fool GNN-based spatiotemporal forecasting models because of their causality and spatiotemporal mechanism, which is detailed in Section 2.

The goal of this paper is to understand and examine the vulnerability and robustness of GNN-based spatiotemporal forecasting models. In doing so, we design a One Vertex Attack (OVA) framework

to break these forecasting models by manipulating only one vertex in the graph. We first propose a universal attack method against spatiotemporal GNNs by applying the inverse estimation to avoid using future ground truth. Then, we utilize the genetic algorithm, whose evaluation function is composed of the proposed universal attack method, to locate the "weakest" vertex. Here the weakest vertex refers to the vertex where attacking it will cause maximum damage to the forecasting models. Finally, we generate perturbations by solving an optimization problem.

It should be noted that poisoning all vertices even multiple vertices in real-world applications is impossible, because the large scale of graph. For instance, the graph of traffic forecasting applications generally covers 1000 square kilometers, and it is unrealistic to organize Harker vehicles to poison all vertices in such a large scale road network. Hence, the proposed one-vertex attack is a realistic solution to evaluate the robustness and vulnerability of spatiotemporal forecasting models deployed in real-world applications.

To prove the effectiveness of the proposed OVA method, we test it in two spatiotemporal traffic datasets with three different Spatiotemporal GNNs. The proposed method can cause at least 15% accuracy drop, and there are about 10% vertices severely impacted with the boundary of speed variation limited to 15km/h.

The contribution of this paper can be summarized as follows.

- First, to the best of our knowledge, this is the first study on attacking Spatiotemporal GNNs by poisoning only one vertex.
- Second, we proposed a novel OVA method that is able to find the weakest vertex and generate optimal adversarial perturbations.
- Third, we empirically study the effectiveness of the proposed method with multiple experiments on real-world datasets.

## 2 RELATED WORK

**Adversarial Attacks against Time Series Analysis.** Some previous works (Chen et al., 2019; Zhou et al., 2019; Alfeld et al., 2016; Karim et al., 2019) proposed adversarial attack methods against Autoregressive models or time series classification models. The above works only consider univariate time series. Different from these works, we focus on complex spatiotemporal domains. The input of spatiotemporal GNNs is the temporal dynamic graph rather than regular matrices or sequences. We take the spatial correlation into consideration while the above works didn't.

**Adversarial Attacks against Graph Neural Networks.** Many studies (Dai et al., 2018; Zugner & Gunnemann, 2019; Chang et al., 2020; Tang et al., 2020) utilized Reinforcement Learning (RL), meta learning, or genetic algorithm to fool GNNs in node, edge, and graph classification domains by tuning the graph topology. All these studies involve no temporal variation in their graphs, and they mainly focus on the spatial pattern. These cannot be applied to fool spatiotemporal forecasting models because of the lack of temporal correlation. Particularly, attacking spatiotemporal forecasing models deployed in real-world applications by graph topology-based attack methods (Zugner & Gunnemann, 2019; Chang et al., 2020) are unrealistic, because tuning the graph topology represents tuning the sensor network that collects spatiotemporal data continuously and any modification on sensors can be easily sensed by the sensor network manager.

**Adversarial Attacks against Recurrent Neural Network.** Recent studies (Rosenberg et al., 2019; Papernot et al., 2016b; Hu & Tan, 2017) demonstrated RNN classifiers were vulnerable to adversarial sequences. These adversarial works require the ground truth to compute adversarial sequences. Because of the forecasting applications' causality, the future ground truth is unavailable. Besides, these works focus on regular vectors or matrices, rather than irregular graphs. Hence these adversarial sequence generation models cannot be directly applied to attack spatiotemporal GNN-based forecasting models.

**One Pixel Attack for Fooling Deep Neural Networks.** Su et al. (2019) utilized Differential Evolution (DE) to generate the perturbation to poison one pixel in images, and then fool CNNs. Similar to one pixel attack, we only poison one vertex in graphs. However, images are regular-structured,

and Su et al. (2019) consider no temporal variation. In addition, one pixel attack requires the ground truth to compute perturbations. In forecasting applications, the ground truth is the future traffic state, and it is inaccessible. The above features prevents one pixel attack from poisoning spatiotemporal forecasting models.

## 3 METHODOLOGY

### 3.1 SPATIOTEMPORAL SEQUENCE FORECASTING AND SPATIOTEMPORAL GNNS

Because of the impossibility of deploying sensors as a regular grid in real-world applications, the form of spatiotemporal data is generally irregular. Consequently, to better mine the spatial information, the spatiotemporal sequence is represented as a temporally varied graph rather than a regular grid. The spatiotemporal sequence can be represented as $\mathcal{G}_t = \{\mathcal{V}_t, \mathcal{E}, W\}$, where $\mathcal{E}$ is the set of edges in the graph, $\mathcal{W}$ is the weighted adjacency matrix whose every element describe the spatial relationship between different variates, $\mathcal{V}_t = \{v_{1,t}, \ldots, v_{n,t}\}$ is the set of condition values (e.g. traffic speed and traffic volume) collected from sensors on timestamp $t$, and $n$ is the number of sensors (Shuman et al., 2013).

Multistep spatiotemporal sequence forecasting can be formulated as Equation 1. Previous conditions from timestamp $t - N + 1$ to $t$ are fed into a forecasting model $F$ that outputs predictions of future conditions from $t + 1$ to $t + M$. In general, $M \leq N$. The above process is customarily called sequence-to-sequence (seq2seq) forecasting.

$$\{\mathcal{G}_{t+M}^*, ..., \mathcal{G}_{t+1}^*\} = F(\{\mathcal{G}_t, ..., \mathcal{G}_{t-N+1}\}) \tag{1}$$

where $\mathcal{G}_i^*$ denotes the prediction of the condition on timestamp $i$.

Most state-of-art spatiotemporal sequence forecasting models output a single future condition, which will be in turn fed as input into the model to forecast the next condition. This process is named as the recursive multistep forecasting, which can be represented as Equation 2.

$$\begin{cases} \mathcal{G}_{t+1}^* = F(\{\mathcal{G}_t, \mathcal{G}_{t-1}, ..., \mathcal{G}_{t-N+1}\}) \\ \mathcal{G}_{t+2}^* = F(\{\mathcal{G}_{t+1}^*, \mathcal{G}_t, ..., \mathcal{G}_{t-N+2}\}) \\ \qquad \vdots \\ \mathcal{G}_{t+M}^* = F(\{\mathcal{G}_{t+M-1}^*, \mathcal{G}_{t+M-2}^*, ..., \mathcal{G}_{t-N+M}\}) \end{cases} \tag{2}$$

Most state-of-art forecasting models, $F$, are constructed based on spatiotemporal GNNs (Li et al., 2017; Wu et al., 2019; Yu et al., 2018; Guo et al., 2019). Spatiotemporal GNNs are composed of both spatial layers and temporal layers. In general, gated linear unit (GLU) or Gated-CNN (Bai et al., 2018) works as the temporal layer to capture the temporal patterns embedded in the spatiotemporal sequence, and the Graph-CNN (Shuman et al., 2013; Bruna et al., 2014) works as spatial layers to capture the spatial patterns.

In this paper, we focus on adversarial studies towards recursive multistep spatiotemporal sequence forecasting. Our studies can be easily extended to seq2seq multistep forecasting.

### 3.2 UNIVERSAL ADVERSARIAL ATTACK AGAINST SPATIOTEMPORAL GNNS

In this section, we point out the form of adversarial attack against the spatiotemporal forecasting, and outline the gap between attacking spatiotemporal GNNs and attacking CNNs or GNNs. Then we propose the inverse estimation to fill the gap. Finally, we design the universal adversarial attack against spatiotemporal GNNs.

#### 3.2.1 ADVERSARIAL ATTACKS AGAINST SPATIOTEMPORAL FORECASTING

Adversarial attacking against recursive multistep forecasting can be formed as Equation 3. The goal is to mislead spatiotemporal GNNs to generate false forecasting by adding perturbations.

$$F(\{\mathcal{G}_t, ..., \mathcal{G}_{t-N+1}\} + \{\rho_t, ..., \rho_{t-N+1}\}) \neq \mathcal{G}_{t+1}$$
$$s.t. \ \|\rho_i\|_p \leq \xi \ \forall \ i \in \{t, ...t - N + 1\} \tag{3}$$

where $\rho_i$ denotes perturbations on timestamp $i$, $\|\cdot\|_p$ denotes $\ell_p$-norm, and $\xi$ denotes the pre-defined constant to constrain the perturbation scale. In real-world traffic applications, $\xi$ control the hacker's driving behavior to balance the attack performance and detection avoidance.

Because of spatiotemporal sequence's causality, we cannot access the future condition that works as the ground truth of forecasting models. In other words, $\mathcal{G}_{t+1}$ in Equation 3 is not available on timestamp $t$. Previous adversarial studies against CNNs, RNNs, and GNNs (Dai et al., 2018; Papernot et al., 2016b; Kurakin et al., 2016b; Alfeld et al., 2016) almost all involve the ground truth in the perturbation computation. In fooling spatiotemporal GNNs as Equation 3, the ground truth, $\mathcal{G}_{t+1}$, is still inevitable. As we mentioned above, the future condition is unavailable, and thus we cannot generate adversarial perturbations directly as Equation 3.

### 3.2.2 INVERSE ESTIMATION

We propose Inverse Estimation to avoid using the future ground truth in fooling spatiotemporal GNNs. First, Equation 3 is transformed to Equation 4, which represents our goal is to fool spatiotemporal GNNs to generate opposite forecasting.

$$\underset{\{\rho_t, ..., \rho_{t-N+1}\}}{\arg\min} \|F(\{\mathcal{G}_t, ..., \mathcal{G}_{t-N+1}\} + \{\rho_t, ..., \rho_{t-N+1}\}) - \tilde{\mathcal{G}}_{t+1}\|_2 + \alpha \cdot \sum_{i=t-N+1}^{t} \max(0, \rho_i^2 - \xi) \tag{4}$$

where $\tilde{\mathcal{G}}_{t+1}$ denotes the opposite condition of $\mathcal{G}_{t+1}$, $\alpha$ denotes the penalty factor. The constrain in Equation 3 is replaced with a regularization term in Equation 4 to constrain the perturbation scale. The penalty factor $\alpha$ is set as 100 to make sure the scale penalty term is much larger than the first term in Equation 4 so that the scale of the computed perturbation is strictly forced. The above idea is similar to targeted attacks (Akhtar & Mian, 2018). However, classical targeted attacks still utilize the ground truth in perturbation computations.

To use no future information, the opposite of future condition, $\tilde{\mathcal{G}}_{t+1}$, is estimated by computing the opposite of the most recent condition, which is represented as Equation 5.

$$\tilde{\mathcal{G}}_{t+1} \leftarrow \tilde{\mathcal{G}}_t = \{\tilde{\mathcal{V}}_t, \mathcal{E}, W\} \tag{5}$$

where $\tilde{\mathcal{V}}_t = \{\tilde{v}_{1,t}, ..., \tilde{v}_{n,t}\}$ denotes a collection of condition values opposite to these collected from sensors. Take the traffic condition for instance, when the condition is "congested/low speed", its opposite is "free/high speed", and vice versa. $\tilde{v}_{i,t}$, the opposite of $v_{i,t}$, is computed as Equation 6.

$$\tilde{v}_{i,t} = \begin{cases} \max(\mathcal{V}), \ v_{i,t} < mid \\ \min(\mathcal{V}), \ v_{i,t} \geq mid \end{cases} \tag{6}$$

where $mid$, $max(\mathcal{V})$, and $min(\mathcal{V})$ represent the mean, maximum, and minimum value of the spatiotemporal dataset, respectively.

Table 1: Comparison on different estimation strategies.

| strategy | PeMS | | | |
| --- | --- | --- | --- | --- |
| | MAE | MAPE (%) | RMSE | PER(%) |
| MR | 1.48 | 3.30 | 2.40 | 13.33 |
| STGCN | 1.39 | 3.06 | 2.37 | 10.55 |
| ARIMA | 1.92 | 3.64 | 3.02 | 12.17 |
| IE | **0.56** | **0.99** | **1.63** | **99.56** |

Inverse Estimation outperforms directly estimating the future ground truth. The error of the estimation on the opposite of ground truth is smaller than errors of any direct estimation. To validate the above assumption, we carry out a test on PeMS dataset. We compare the proposed inverse estimation with three types of ground truth estimation methods, namely estimating by the most recent traffic condition (MR), spatiotemporal graph convolution neural network (STGCN), and AutoRegressive Integrated Moving Average (ARIMA). The experiment result is shown as Table 1. The proposed Inverse Estimation's performance, including mean absolute error (MAE), mean absolute percent error (MAPE), root mean square error (RMSE), and perfect estimation ratio (PER), is better than others'. It should be noted 99.56% IE's estimations are exactly equal to the opposite of ground truth.

### 3.2.3 UNIVERSAL ATTACKS AGAINST SPATIOTEMPORAL SEQUENCE FORECASTING

Adversarial perturbations generated by the subsection 3.2.2 vary with the input graph. Only if the perturbation keeps being updated, it will be effective all the time. The universal attack denotes that the perturbation is consistent and independent from the input, which can be represented as Equation 7.

$$\arg \min_{\rho_u} \|F(\{\mathcal{G}_t, ..., \mathcal{G}_{t-N+1}\} + \{\rho_u\}) - \tilde{\mathcal{G}}_{t+1}\|_2 + \alpha \cdot max(0, \rho_u^2 - \xi) \qquad (7)$$

where $\rho_u$ denotes the universal perturbation.

The universal perturbation can be generated by solving equation 7. After the universal perturbation is generated, there is no need to update it when new data come. The proposed universal attack will be utilized to locate which vertex to attack for one vertex attack.

### 3.3 LOCATING WEAKEST VERTEX

In this subsection, we first mathematically define the "weakness" of a vertex in a graph. The "weakness" of the $jth$ vertex denotes the number of influenced vertices when the $jth$ vertex is attacked by the proposed universal universal perturbation, which is shown as equation 8.

$$weak_j = \|K_\theta \left\{ F(\{\mathcal{G}_t, ..., \mathcal{G}_{t-N+1}\} + M_j \cdot \rho_u) - \mathcal{G}_{t+1} \right\}\|_0 \qquad (8)$$

where $M_j \cdot \rho_u$ denotes that all elements except the one corresponding $jth$ vertex are set to 0, and $K_\theta\{\cdot\}$ denotes an element-wise filter to set elements whose absolute value is smaller than $\theta$ to 0. A greater "weakness" value represents more vertices will be influenced if the $jth$ vertex is attacked. We will attack the vertex with the largest "weakness" value. In traffic forecasting applications, $\theta$ is set as 5 empirically.

A possible method to locate the weakest vertex is the complete traversal algorithm. However, this method is time consuming. To reduce the time cost, we utilize the genetic algorithm to locate the weakest vertex, which is shown as follows.

- First, the initial candidate set is composed of $s$ vertices with the most edges.
- Second, the updated candidate set is computed as equation 9

$$v_i(g+1) = v_{r1}(g) + p(v_{r2}(g) - v_{r3}(g)) \qquad (9)$$

  where $v_i$ denotes the position of the $ith$ vertex, $g$ denotes the $gth$ iteration, $r1$, $r2$, and $r3$ are random numbers with different values, and $p$ is the parameter set to be 0.5 empirically.

- Third, compare the updated candidates' weakness with the previous candidates', then keep $s$ candidates with the largest weakness value.

- Fourth, repeat the second and third step until the candidate set is consistent or $g > 10$. Select the weakest vertex to attack. It should be noted that the bound of $g$ controls the trade-off of the proposed solution's effectiveness and efficiency. The larger bound represents the proposed solution is much closer to the complete traversal algorithm.

### 3.4 ONE VERTEX ATTACKING AGAINST SPATIOTEMPORAL GNNS

After the weakest vertex is located, we generate one vertex perturbation based on equation 10. Poisoning the weakest vertex in a graph with the carefully designed perturbation can fool the spatiotemporal GNN-based traffic forecasting system.

$$\underset{M_J \cdot \{\rho_t, ..., \rho_{t-N+1}\}}{\arg\min} \|F(\{\mathcal{G}_t, ..., \mathcal{G}_{t-N+1}\} + M_J \cdot \{\rho_t, ..., \rho_{t-N+1}\}) - \tilde{\mathcal{G}}_t\|_2 + \alpha \cdot \mathcal{R}_{onevertex}$$

$$\mathcal{R}_{onevertex} = \sum_{i=t-N+1}^{t} max(0, (M_J \cdot \rho_i)^2 - \xi) \tag{10}$$

where $J$ denotes the index of the weakest vertex, and $M_J \cdot \rho_i$ is the generated perturbation. It should be noted that $\|M_J \cdot \rho_i\|_0 \leq 1$.

Different from adversarial attack methods proposed as Equation 4 and Equation 7, the one vertex attack keeps poisoning one vertex in the graph, while others poison the entire graph. As for real-world traffic forecasting applications, poisoning the entire sensor network deployed in road networks is unrealistic, while one vertex attack is much more harmful.

In reality, the vertex denotes a sensor like a loop detector. The one vertex perturbation denotes a vehicle's speed shift. If a hacking vehicle's speed varies following the perturbation computed as Equation 10, it can fool the entire traffic forecasting system, not only at the vertex where the hacking vehicle is, but also at other vertices and even vertices far away from the attacked vertex.

## 4 EVALUATION AND RESULTS

The evaluation of the proposed method is based on two traffic datasets, namely PeMS and METR-LA(S). PeMS records 44-days traffic speed data which was collected from 200 stations of Caltrans Performance Measurement System (PeMS). METR-LA(S) records four months of traffic speed on 100 stations on the highways of Los Angeles County. Our experiments are conducted under an NVIDIA DGX station with 4 NVIDIA Tesla V100 GPU cards.

We test three spatiotemporal GNNs including STGCN (Yu et al., 2018), DCRNN (Li et al., 2017), and Graph WaveNet (Wu et al., 2019). Each dataset is split into 3 subsets: 70% for training, 10% for validation, and 20% for testing. All setting parameters are as same as related papers (Yu et al., 2018; Li et al., 2017; Wu et al., 2019) except the number of input and output channels accordingly set as the number of stations in the said dataset. In addition, we use the validation set to locate the attack position, and generate OVA perturbations in real-time for the test set. As for measurement, we introduce three metrics to measure the proposed method's effectiveness.

- **MAPE Increase (MAPEI)** - Mean Absolute Percentage Error (MAPE) is a measure of prediction accuracy and smaller MAPE represents better predictions. An increase in MAPE thus translates into a decrease in the prediction accuracy.

- **Normalized MAPE Increase (NMAPEI)** - It denotes the ratio between MAPEI and the MAPE before attacking.

- $k\%$-**Impacted Vertices** - Counts the number of vertices with NMAPEI being greater than $k\%$.

### 4.1 TRADEOFF BETWEEN ATTACK PERFORMANCE AND DETECTION AVOIDANCE

In real-world traffic applications, the generated perturbations represent the hacking vehicle's speed shifts. The parameter $\xi$, which is used to limit the driving behavior, in Equation 10 balances the attack performance and detection avoidance.

We first propose an experiment to test how the parameter $\xi$ influences the effectiveness of the proposed one vertex attack method. In this subsection, 15min traffic speed forecasting is undertaken by STGCN, DCRNN, and Graph Wavenet that work as the targeted models and the experiment is conducted on META-LA(S). These models are attacked by the proposed OVA with different $\xi$.

Table 2 shows the number of impacted vertices with different $\xi$. When the $\sqrt{\xi}$ is equal to 20, over 10% vertices, whose NMAPEI are greater than 40%, are severely impacted, even with only one vertex attacked. With a small $\sqrt{\xi}$, there are about 50% vertices are influenced. Based on the results shown as Table 2, we can conclude that perturbations will be diffused from one vertex to most of the graph when we apply spatiotemporal GNN-based forecasting models. The greater the perturbation is, the more vertices in the graph will be influenced.

Table 2: The relationship between $\sqrt{\xi}$ and the $k\%$-Impacted Vertices

| $\sqrt{\xi}$ $k\%-$ | STGCN | | | | DCRNN | | | | Grave Wavenet | | | |
|---|---|---|---|---|---|---|---|---|---|---|---|---|
| | 5 | 10 | 15 | 20 | 5 | 10 | 15 | 20 | 5 | 10 | 15 | 20 |
| 5%-Impacted Vertices | 43 | 71 | 69 | 90 | 51 | 75 | 82 | 88 | 42 | 70 | 82 | 91 |
| 10%-Impacted Vertices | 14 | 52 | 61 | 82 | 17 | 59 | 65 | 71 | 16 | 32 | 52 | 73 |
| 20%-Impacted Vertices | 4 | 22 | 40 | 46 | 8 | 27 | 45 | 50 | 3 | 25 | 41 | 55 |
| 30%-Impacted Vertices | 0 | 1 | 17 | 39 | 1 | 1 | 26 | 40 | 1 | 4 | 19 | 38 |
| 40%-Impacted Vertices | 0 | 1 | 1 | 9 | 0 | 0 | 0 | 13 | 0 | 0 | 1 | 16 |

Setting $\xi$ into an appropriate range is important. An extremely large $\xi$, which represents abnormal driving behaviors in traffic domains, will be detected easily. By analyzing PeMS and META-LA(S), speed variation within 15km/h occurs frequently, and consequently, we regard the accessible boundary of speed variation is 15km/h, namely $\sqrt{\xi} = 15$.

## 4.2 EFFECTIVENESS AND EFFICIENCY OF LOCATING WEAKEST VERTEX

In this subsection, experiments on PeMS are carried out to prove the effectiveness and efficiency of the proposed weakest vertex locating strategy. STGCN works as the model to attack. Three locating strategies, namely locating the vertex with the most edges (MOS), locating the vertex with the highest centrality (CEN), and locating the weakest vertex by the complete traversal algorithm (CT), work as baselines. After locating the weakest vertex by different strategies, perturbations are computed as Equation 10, and then fed into STGCN. NMAPEI and 30%-IV are recorded in Table 3.

Table 3: The effectiveness test on the proposed weakest vertex locating strategy

| | NMAPEI | 30%-IV |
|---|---|---|
| MOS | 4.5% | 3 |
| CEN | 3.2% | 1 |
| CT | **15.2%** | **17** |
| Proposed | **15.2%** | **17** |

The proposed strategy's effectiveness is close to the complete traversal algorithm. In this experiment, it locates the same weakest vertex as the complete traversal algorithm does. Poisoning the vertex with the most edges or the highest centrality cannot fool the forecasting model effectively. A possible reason is that these vertices' robustness is improved by their neighbors because of the STGCN's spatiotemporal mechanism.

In addition, the proposed strategy spends 1104 seconds to locate the vertex to attack, while the complete traversal algorithm spends 1795 seconds. The proposed strategy can reduce the time cost of locating the weakest vertex.

## 4.3 EFFECTIVENESS OF ONE VERTEX ATTACK

In this subsection, experiments on PeMS are designed to prove the effectiveness of the proposed method. 15min traffic speed forecasting is undertaken by three spatiotemporal GNN methods that work as models to attack. We compare the proposed OVA with four baselines that are detailed as follows.

- RAN: Generate perturbations based on Equation 4 and randomly attack one vertex; repeat the above step for 10 times and compute the average MAPEI and NMAPEI.

- RAN2: Generate Gaussian White Noise (GWN) with the same scale constrain as the proposed OVA method shown as Equation 10, and attack the weakest vertex following the proposed weakest vertex locating strategy.

- MOS: Regard the vertex with the most edges in the graph as the weakest vertex, and attack this vertex based on Equation 10.

- MFSM: Attack all vertices with modified Fast Gradient Sign Method (FGSM) (Szegedy et al., 2015). We replace the ground truth in the original FGSM with the proposed inverse estimation as Equation 5, which is shown as Equation 11.

$$\{\rho_t, ..., \rho_{t-N+1}\} = \epsilon \, sign(\bigtriangledown \mathcal{J}(\Phi, \{\mathcal{G}_t, ..., \mathcal{G}_{t-N+1}\}, \tilde{\mathcal{G}}_t)) \tag{11}$$

where $\bigtriangledown \mathcal{J}$ computes the gradient of the cost function around the prediction of the forecasting model parameterized by $\Phi$ w.r.t the input sequence $\{\mathcal{G}_t, ..., \mathcal{G}_{t-N+1}\}$, $sign$ denotes the sign function, $\tilde{\mathcal{G}}_t$ denotes the inverse estimation of the ground truth, and $\epsilon$ control the scale of the perturbations.

Table 4: Effectiveness evaluation based on PeMS

|  | STGCN | | DCRNN | | Grave Wavenet | |
| --- | --- | --- | --- | --- | --- | --- |
|  | NMAPEI | 30%-IV | NMAPEI | 30%-IV | NMAPEI | 30%-IV |
| Proposed | **15.2%** | **17** | **16.7%** | **22** | **15.5%** | **21** |
| RAN | 2.1% | 1 | 2.7% | 0 | 2.3% | 0 |
| RAN2 | 1.7% | 0 | 2.3% | 0 | 2.1% | 0 |
| MOS | 4.5% | 3 | 4.7% | 3 | 5.7% | 2 |
| MFGSM-3 | 27.3% | - | 24.4% | - | 25.8% | - |
| MFGSM-2 | 15.4% | - | 15.6% | - | 16.2% | - |

In these experiments, $\sqrt{\xi}$ is set to 15 for methods that attack only one vertex. "MFGSM-$\epsilon$" is used to point out the perturbation constrain of MFGSM. Because it attacks all vertices rather than one vertex, we set $\epsilon$ as 3 and 2 respectively.

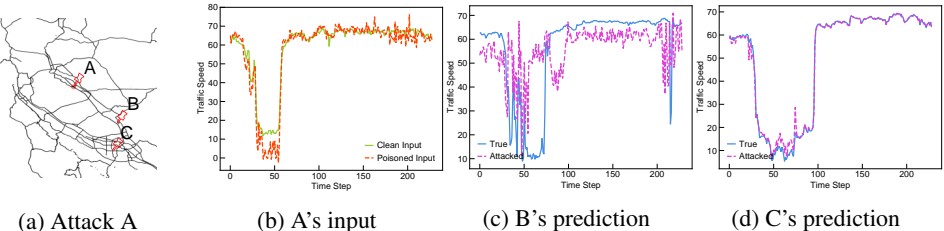

(a) Attack A  (b) A's input  (c) B's prediction  (d) C's prediction

Figure 1: Results of the proposed One Vertex Attack

Table 4 shows the experiment results. Our proposed one vertex attack method outperforms attacking one vertex randomly (RAN) and attacking the vertex with the most edges (MOS), which represents that the proposed method on locating the weakest vertex works. Our method outperforms attacking the weakest vertex with GWN (RAN2), which represents the proposed method can generate the optimal perturbations for one vertex attack.

Fig 1a shows an attack result on STGCN. Vertex A is attacked with the poisoned input as Fig 1b, and predicted sequences in vertex B and C are shown as Fig 1c and Fig 1d respectively. B and C are far away from A and there is no attack on B and C. Fig 1c shows an example that the spatiotemporal GNN-based forecasting model mispredicts "congestion" as "uncongested" at about the 60th step.

For traffic forecasting in PeMS, the proposed method can cause over 15% accuracy drop for all three Spatiotemporal GNN models, and about 10% vertices are seriously impacted (these vertices' NMAPEI are greater than 30%) with $\sqrt{\xi}$ equal to 15.

Attacking all vertices are always better than attacking only one vertex, but the proposed method's effectiveness is similar with attacking all vertices by MFSDM with $\epsilon$ approximately equal to $10\% \cdot \sqrt{\xi}$, which is concluded by comparing the proposed method with MFGSM-3 and MFGSM-2 in table 4. It should be noted that MFGSM attacks 200 vertices, while OVA only attacks one.

## 5 CONCLUSION

This paper proposed One Vertex Attack that can break spatiotemporal GNN-based forecasting models by poisoning only one vertex. The generated perturbation will be diffused to numerous vertices in the graph when Spatiotemporal GNNs are under attack.

Future works for the proposed study can be summarized as follows. First, we utilized a universal adversarial attack method to measure the "weakness" of vertices. We do not include temporal patterns in our measurement. Consequently, involving temporal patterns in the evaluation is a possible modification. Second, we design a genetic algorithm-based method to find the "weakest" vertex in a graph to attack. This might not be the optimal solution. Third, studies on the scalability of one vertex attack is valuable.

Besides, as spatiotemporal applications require reliable algorithms, how to defend these adversarial attacks, and how to build more robust spatiotemporal GNN-based models are still valuable.

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
