# OpenReview forum: "One Vertex Attack on Graph Neural Networks-based Spatiotemporal Forecasting"
_ICLR.cc/2021/Conference — Reject_

### Official Review · AnonReviewer1 · 2020-10-27
**This paper proposes a single vertex based white-box attack on spatiotemporal GNNs.**

**Rating:** 4
**Confidence:** 3

**Review:**

This paper proposes a single vertex based white-box attack on spatiotemporal GNNs. They solve the optimization of the least perturbations with the inverse estimation. The results demonstrate that the perturbation on one vertex can mislead the performance of the GNNs.

Advantages:
The idea of designing a single one vertex based attack is novel; the experiments are exhaustive, and the results validate the effectiveness of the attack method.

Concerns:
I think the technical contribution of this paper is weak in that:
1. One-pixel attack [1] is not new, I don’t see some novelties in designing the one vertex attack. The only interesting point is the expansion into the new scene of attacking GNNs.
2. I'm wondering the efficiency of the one vertex attack. Is it really important to implement the one vertex attack? Besides, how's the attack success rate if the STGCN, DCRNN and Graph WaveNet are all aware of the attack method, i.e. the performance of attacks on defensed neural networks? For example, preprocessing is a routine method for defending against adversarial attacks. If the denfense side is aware of the attack, then how's the attack success rate when the attacked model is equipped with a simple filtering operation before the test sample are fed into the GNN?
3. As far as I'm concerned, the adversarial attack and defense are the arms race, I'm not surprised the one vertex attack has a strong attack success rate since it is a white-box attacked and it computes the modification by solving an optimization problem. What really matters is why it is important to attack a GNN like network and what's the novelty, since we all know that deep neural network can be easily attacked by adversarial examples. I wish to see the real discoveries rather than just adapting an attack mechanism to a new classification task.

[1] Su J, Vargas D V, Sakurai K. One pixel attack for fooling deep neural networks[J]. IEEE Transactions on Evolutionary Computation, 2019, 23(5): 828-841.

---

> ### Author Response · Authors · 2020-11-25
> **The proposed method is the first adversarial attack method that can evaluate the spatiotemporal forecasting models in real-world applications**
>
> Dear Reviewer1,
>
> 1.
> Our proposed method is totally different from the One-pixel attack, and the difference had already been discussed in the fourth paragraph of related work.
>
> As we had mentioned, one-pixel attack cannot be applied to attack the spatiotemporal GNN. First, one-pixel attack relies on the ground truth to compute the perturbations, while the ground truth is not available in spatiotemporal forecasting applications. Second, one-pixel attack focuses on poisoning images that are regular matrice, and it does not consider any temporal patterns and correlations. For spatiotemporal forecasting applications, the data structure is the temporally dynamic graph.
>
> All in all, one vertex attack and one-pixel attack are two different methods targeting two different applications.
>
> 2.
> It is extremely important to implement the one vertex attack to evaluate the robustness and vulnerability of real-world spatiotemporal forecasting models.
>
> As we had already mentioned in the second paragraph in section 3.4, all attacking strategies that poison all vertice are unrealistic in real-world applications.
>
> Take traffic forecasting for example, the entire graph consists of hundreds even thousands of vertice that are loop detectors deployed in real-world road networks. The entire graph covers almost 1000 square kilometers. It is impossible to poison all sensors at the same time.
>
> Spatiotemporal forecasting models’ vulnerability cannot be evaluated by any previous adversarial attack studies. To make up the gap, we propose one vertex attack, that can break the forecasting model by poison only one vertex in real-world applications.
>
> 3.
> As we had clearly mentioned in our title, abstract, and introduction, we focused on the spatiotemporal forecasting models. Forecasting is not a classification task.
>
> As we had discussed in related work, previous studies cannot be applied to attack spatiotemporal forecasting models. There are two main reasons. (1), most previous works utilize the ground truth to compute the perturbation. For instance, the one-pixel attack paper you mentioned involves the image label in the perturbation computation. However, for the spatiotemporal forecasting application, the ground truth is not accessible. You can never know the true state at 3 pm when the current time is 2 pm. (2) previous works do not consider the spatiotemporal graph that consists of both spatial and temporal patterns.
>
> We all know deep learning is vulnerable and deep learning, especially spatiotemporal GNNs, are gradually applied to address real-world forecasting problems. The gap is that previous adversarial works cannot be applied to evaluate these forecasting models’ robustness and vulnerability because of spatiotemporal structure and inaccessible ground truth. Our real discovery is that the proposed method is the first method that addresses these issues and can evaluate the forecasting models in real-world applications.

---

### Official Review · AnonReviewer2 · 2020-10-28
**A simple and effective one vertex attack method on graph neural networks for spatio-temporal prediction problems**

**Rating:** 4
**Confidence:** 3

**Review:**

This paper studies the problem of attacking graph neural networks for spatio-temporal prediction problems (e.g., traffic speed prediction). The input of the problem is a spatio-temporal sequence represented as graphs at time t-N+1 to t, where a graph neural network is trained to predict the graph sequence for time t+1 to t+M. The aim is to add perturbations to the input graph sequence, such that the predicted graph sequence varies from the ground truth as much as possible.

The novelty of the proposed attack model lies in that it does not require knowledge on the ground truth graph sequence. Instead, it uses an inverse estimation that uses the "opposite" of the graph at time t as the perturbation target.

For efficiency and practicality considerations, the vertex that has the maximum impact on the prediction output is computed using a heuristic algorithm, and perturbation is done on this vertex only. This makes the attack more realistic as only one vertex needs to be attacked rather than the full graph.

Pros:
1. The problem studied is interesting. Spatio-temporal prediction problems have many applications, and many recent works on such problems do use graph neural networks to make predictions.

2. Experimental results on real datasets show that the proposed algorithms are effective and efficient.

Cons:
1. The proposed models and algorithms are shown to be effective. One concern is on the technical depth. The modelling of the problem and the algorithms are relatively straightforward. There is a lack of performance guarantees and theoretical guidance on the choice of parameter values. For example, how much worse in terms of the attack impact when the weakest vertex is found by the heuristic algorithm instead of by a full traversal? Why is g bounded by 10 in Equation 9? Are there guidance on choosing \alpha in Equation 4 beyond "empirically"? What is the relationship between the number of vertices attacked and the impact of the attack?

Additional comments:
How exactly are the experiments in Table 1 set up? Are all algorithms/models in Table 1 given the same input and ground truth output for the error calculation? If not (which seems to be the case based on the current wording), the comparison might be unfair.

What are the meanings of "position" of a vertex and '-' in Equation 9? What is the intuitive idea that the genetic algorithm suits the problem to find the weakest vertex?

What exactly does the right arrow in Equation 5 mean? Assigning \Tilde{g}_t to \Tilde{g}_{t+1}?

The related work section claims that "these works (i.e., works on adversarial attacks against recurrent neural network) focus on regular vectors or matrices, rather than irregular graphs." It would be good to clarify the difference between "regular matrices" and "irregular graphs", since graphs are often represented by their adjacency (or vertex feature) matrices.

Typo: Check the double quotes throughout the paper.

**Update after author response:** I appreciate the authors' efforts to address my comments. Part of the comments on the presentation of technical details have been addressed, such as the meaning of "position" of a vertex and "-" in Equation 9. However, the main concern in terms of the technical depth has not been addressed well. There is a lack of performance guarantees and theoretical guidance on the choice of parameter values. For example, how much worse in terms of the attack impact when the weakest vertex is found by the heuristic algorithm instead of by a full traversal? The results in Section 4.2 and Table 3 are empirical and not theoretical. As such, I could not change to a more positive rating.

---

> ### Author Response · Authors · 2020-11-25
> **The depth of the proposed method was discussed in one of experiment parts, section 4.2.**
>
> Dear Reviewer4,
>
> Thanks for your considerable and helpful comments.
>
> 1.
> Section 4.2 and Table 3 had discussed and showed the comparison between the heuristic algorithm and the complete traversal. For PeMS data, the proposed one can find the same weakest vertex as the complete traversal and only use 60% time cost.
>
>
> the penalty \alpha in Equation 4 is set as 100 to make sure the scale of the penalty term is much larger than the scale of the first term so that the scale of perturbations can be strictly forced. There is almost no difference between 100, 1000, …. We add this statement in 3.2.2. Thanks for your comment.
>
> The g in Equation 9 control the tradeoff between effectiveness and efficiency. When g is extremely large, the proposed weakest vertex location solution is as same as the full traversal.
>
> We attack only one pixel because of the real-world constraint. For spatiotemporal forecasting applications, the graph is determined by the sensor network that collects data. Take traffic forecasting for instance, the graph covers almost 1000 km2. Supposing vertex A and vertex are not neighbors, the distance between A and B might be 100km. It is complicated to attack A and B by the proposed one-vertex attack algorithm at the same time with different Harker vehicles in the real world (because the traffic flows in A and B are generally different). So we attack only one vertex to evaluate the spatiotemporal forecasting models. Attacking multiple vertices should be developed in future work.
>
> 2.
> Table 1’s experiment is set up as the same as the experiment section. Table 1 does use different input and ground truth. But errors calculated and compared in Table 1 are the exact errors that will be diffused into Equation (4) (7) (8). Table 1 clearly shows that the proposed inverse estimation can involve much less error in the perturbation computation, compared with the baselines listed in Table 1.
>
> 3.
> “position” of a vertex represents the real-world coordinate of the vertex. We target real-world forecasting applications. Our graph comes from the real-world sensor network that collects data continuously. Each sensor is a vertex, and it has its own coordinate on earth.
>
> “-” in Equation 9 represents the subtraction between different coordinates.
>
> Candidates of the weakest node can be selected by a reward function. An iterative and time-saving solution is the genetic algorithm. It should be noted that the proposed genetic algorithm might not be the optimal one to locate the weakest vertex. The proposed one had outperformed other possible solutions as we had discussed in Section 4.2.
>
> 4.
> Yes, it is assigning. We replace Equation 5 with  \Tilde{g}{t+1}<-\Tilde{g}t , which might be easy to understand.
>
> 5.
> One matrix cannot represent a graph completely. A graph should be represented by two matrices at least. One matrix is the adjacency matrix that consists of the spatial correlation between different vertice. The other matrix is the vertex feature matrix. Moreover, we consider the spatiotemporal graph here. The feature matrix should be a tensor (multiple temporally dynamic matrix).  State-of-art spatiotemporal forecasting models [1][2][3][4] stated the difference between the normal matrix and graph. That is also the reason why these forecasting models outperform LSTM or GRU, because LSTM and GRU can only utilize part of the graph (the feature matrix only).
>
>
> [1]Shengnan Guo, Youfang Lin, Ning Feng, Chao Song, and Huaiyu Wan. Attention based spatialtemporal graph convolutional networks for traffic flow forecasting. In AAAI, 2019.
>
> [2]Yaguang Li, Rose Yu, Cyrus Shahabi, and Yan Liu. Diffusion convolutional recurrent neural network: Data-driven traffic forecasting. arXiv preprint arXiv:1707.01926, 2017.
>
> [3]Bing Yu, Haoteng Yin, and Zhanxing Zhu. Spatio-temporal graph convolutional networks: A deep learning framework for traffic forecasting. In IJCAI, 2018.
>
> [4]Zonghan Wu, Shirui Pan, Guodong Long, Jing Jiang, and Chengqi Zhang. Graph wavenet for deep spatial-temporal graph modeling. In IJCAI, 2019.

---

### Official Review · AnonReviewer4 · 2020-10-28

**Rating:** 8
**Confidence:** 3

**Review:**

This paper investigates the attack of a spatiotemporal GNN and also proposing a method to find the weakest vertex for this attack. It investigates the strength of the suggested approach. The results are convincing.   It is well written with a few edits to make:
- Page 2, ln 8: Spelling of spatiotemporal
- Page 2, Section 2, par 2: These cannot... instead of These work cannot...
- Page 3, par 2: equation 1 should be repositioned for flow.
- Equation 6: \min etc instead of min etc.
- Page 4, last paragraph: refer to the section by number instead of saying 'above'
- Page 5 and further: weakness and others have the speech commas facing incorrectly outwards. These could also just be removed rather.
- Page 6, par above Table 2: Refers to the 'above results' which are actually below in Table 2 - use labels and ref.
- There are many unpublished references included - these should possibly be reduced - many are probably now published?
- Ref Dai 2018 has et al in it?

---

> ### Author Response · Authors · 2020-11-25
> **We modify our paper following your suggestions.**
>
> Dear Review4,
>
> Thanks so much for your careful reading and helpful comments.
>
> We modify our paper following your suggestions.
>
> Particularly, we add one paragraph (the fifth paragraph) into our introduction to declare why the proposed one-vertex attack is the realistic solution to evaluate the robustness and vulnerability of the spatiotemporal forecasting GNNs, and why poisoning all vertices is unrealistic in the real world.
>
> Besides, we also modify some equations (1) (2) (5) to make them be much easier to understand.
>
> In addition, we add the reason to choose the hyper-parameters.

---

### Official Review · AnonReviewer3 · 2020-10-29
**The attack only focuses on a very small part on the spatiotemporal neural network.**

**Rating:** 4
**Confidence:** 5

**Review:**

The paper proposes a new one vertex adversarial attack to evaluate the robustness on deep spatialtemporal graph neural network. First, it formally defines the adversarial attack into optimization problem and uses a genetic algorithm to solve it.

Pros:
1. To my best of knowledge,  none adversarial attack has been applied to spatialtemporal graph neural network. And the spatialtemporal graph neural network has its uniqueness in terms of unfixed input and graph property.
2. The attack performance seems reasonable.

Cons:
1. While it has proposed a new attack towards the spatialtemporal graph neural network, the novelty of the proposed method is not significant. It could just be treated as attack the node features on a normal graph neural network to predict the next node' label.
2. The difficulty of unfixed or interactive input on spatialtemporal graph neural network is avoided instead of solving it. There are some papers like [1][2] that already attack the interactive input and address the problem instead of avoiding it.
3. The universal attack claimed is not actually a universal attack. The universal attack is defined to use a common perturbation on features on every node instead of just one node. I suggest to take out the universal claims.
4. While attacking the node attributes or features, it would be more interesting to utilize the graph topology as well, which is commonly discussed on the adversarial attacks related to graph neural networks.
5. The constraint on the perturbation is actually a l2 constraint however it is a soft constraint. I am not sure whether the constraint is strictly forced. If not, how to measure the node feature's similarity after the perturbation?


[1] Cheng, Minhao, Wei Wei, and Cho-Jui Hsieh. "Evaluating and enhancing the robustness of dialogue systems: A case study on a negotiation agent." Proceedings of the 2019 Conference of the North American Chapter of the Association for Computational Linguistics: Human Language Technologies, Volume 1 (Long and Short Papers). 2019.

[2] Gleave, Adam, et al. "Adversarial policies: Attacking deep reinforcement learning." arXiv preprint arXiv:1905.10615 (2019).

---

> ### Author Response · Authors · 2020-11-25
> **We proposed the one-vertex attack, that uses no ground truth and breaks forecasting models by poisoning only one vertex, to make up the gap that previous adversarial works cannot be applied to attack real-world deployed spatiotemporal GNNs.**
>
> Dear Reviewer3,
>
> Thanks for your considerable and helpful comments.
>
> 1.
>
> As we mentioned in the related work section, previous GNN adversarial attacks cannot be directly applied to attack spatiotemporal forecasting models because of the inaccessible ground truth. For forecasting applications, we cannot know the true state at 3 pm when the current time is 2 pm, which is called “causality”. As a result, true labels, that are widely used in perturbation computation in most previous adversarial works, cannot be applied in attacking spatiotemporal GNNs. The proposed one-vertex attack addresses the causality issue by the proposed inverse estimation.
>
> 2.
>
> For real-world forecasting applications, the input is generally pre-processed by well-developed solutions[1][2][3] to address the unfixed issues. State-of-art spatiotemporal GNNs forecasting models [4][6][6][7] also do not consider the unfixed issues because they are well addressed in the forecasting domain. The iterative issues are addressed by the spatiotemporal GNN structures.
>
> The proposed method can locate the “weakest” vertex that can break the entire network severely. Attacking different vertice at different timestamps might be unrealistic. Take traffic forecasting for instance, the distance between two vertices (not neighbor vertices) might be over 100km. It is almost impossible to attack vertex A at 2:00 pm and then to attack vertex B that is 100km far away from A at 2:05 pm. Considering the real-world graph may cover almost 1000 square kilometers, we design a strategy that attacks the weakest one vertex to break the entire forecasting model.
>
> 3.
> Our universal attack claim is correct because the proposed universal attack is to attack every node by a common perturbation. The proposed universal attack is applied to locate the weakest vertex, and then the one-vertex attack is applied, which was detailed in section 3.2.3 and 3.3.
>
> 4.
> The graph topology-based attacking is unrealistic for real-world forecasting applications. The graph is determined by the sensor network that collects the data. Manipulating the topology represents ruining the deployed sensor network in the real world. When the sensor is moved, deleted, or added, the manager can easily sense it immediately.
>
> 5.
> The constraint is strictly forced by setting the penalty factor as 100 as we mentioned in 3.2.2. The scale of the second term (the penalty part) in the objective equation (4) (7) (8) is much larger than the first term, which results in keeping the similarity first. Our experiment also shows that the scale of perturbation is strictly forced. Figure 1(b) had shown the similarity between the true input and the poisoned input.
>
> BTW:
>
> We all know deep learning is vulnerable. Deep learning, especially spatiotemporal GNNs, gradually applied in real-world forecasting applications. However, previous adversarial studies cannot directly be applied to attack these spatiotemporal GNNs, because of inaccessible ground truth, spatiotemporal structure, and the large scale of the real-world graph (large scale means attacking the entire network is unrealistic), which was detailed in the related work section. As a result, the robustness and vulnerability of these spatiotemporal forecasting models cannot be evaluated. We proposed the one-vertex attack, that uses no ground truth and breaks forecasting models by poisoning only one vertex, to make up the gap.
>
>
>
>
>
>
> [1] Zhang Z. Missing data imputation: focusing on single imputation. Ann Transl Med. 2016;4(1):9. doi:10.3978/j.issn.2305-5839.2015.12.38.
>
> [2] José J., Ignacio M., Pedro G., Emilio A., Nuria R., Miguel M., and Leonardo F. Missing data imputation using statistical and machine learning methods in a real breast cancer problem. Artificial Intelligence in Medicine. 2010;50(2):105-115. doi:10.1016/j.artmed.2010.05.002.
>
> [3] Yuebia L., Zhiheng L. and Li L. Missing traffic data: comparison of imputation methods.IET Intelligent Transport Systems. 2014;8(1):51-57. dot:10.1049/iet-its.2013.0052.
>
> [4]Shengnan Guo, Youfang Lin, Ning Feng, Chao Song, and Huaiyu Wan. Attention based spatialtemporal graph convolutional networks for traffic flow forecasting. In AAAI, 2019.
>
> [5]Yaguang Li, Rose Yu, Cyrus Shahabi, and Yan Liu. Diffusion convolutional recurrent neural network: Data-driven traffic forecasting. arXiv preprint arXiv:1707.01926, 2017.
>
> [6]Bing Yu, Haoteng Yin, and Zhanxing Zhu. Spatio-temporal graph convolutional networks: A deep learning framework for traffic forecasting. In IJCAI, 2018.
>
> [7]Zonghan Wu, Shirui Pan, Guodong Long, Jing Jiang, and Chengqi Zhang. Graph wavenet for deep spatial-temporal graph modeling. In IJCAI, 2019.

---

### Author Response · Authors · 2020-11-25
**modification illustartion**

Thanks for all reviewers' careful works and considerable comments.

There are some modifications to our paper.

First, we add one paragraph (the fifth paragraph) into the introduction section to declare why poisoning all vertices is an unrealistic solution to evaluate the robustness and vulnerability of spatiotemporal GNNs.

Second, we add the reason to set the penalty factor \alpha for Equation 4 as 100 (section 3.2.2) and the meaning of the bound of iteration of the genetic algorithm (section 3.3). These are about how to set the hyper-parameters.

In addition, we modify Equation (1)(2)(5) to make them easy to understand. The meaning of these equations has not changed anymore.

Finally, there are some grammatical revisions.

Thanks again for your comments.

---

### Decision · Program_Chairs · 2021-01-07
**Final Decision**

**Decision:**

Reject

**Comment:**

This paper proposes one vertex attack for GNN, applied to spatiotemporal forecasting. The paper can be improved w.r.t novelty, incorporating graph topology and rigorous analysis.